# Chimeric Antigen Receptor T Cell and Chimeric Antigen Receptor NK Cell Therapy in Pediatric and Adult High-Grade Glioma—Recent Advances

**DOI:** 10.3390/cancers16030623

**Published:** 2024-01-31

**Authors:** Adrian Kowalczyk, Julia Zarychta, Anna Marszołek, Joanna Zawitkowska, Monika Lejman

**Affiliations:** 1Student Scientific Society of Department of Pediatric Hematology, Oncology and Transplantology, Medical University of Lublin, 20-093 Lublin, Poland; adriankowalczyk31@gmail.com (A.K.); julia.zarychta99@gmail.com (J.Z.); 2Student Scientific Society of Independent Laboratory of Genetic Diagnostics, Medical University of Lublin, 20-093 Lublin, Poland; anna.marszolek@gmail.com; 3Department of Pediatric Hematology, Oncology and Transplantology, Medical University of Lublin, 20-093 Lublin, Poland; joannazawitkowska@umlub.pl; 4Independent Laboratory of Genetic Diagnostics, Medical University of Lublin, 20-093 Lublin, Poland

**Keywords:** CAR-T, CAR-NK, high-grade glioma, tumor immunosuppressive microenvironment

## Abstract

**Simple Summary:**

High-grade gliomas (HGG), both in children and adults, are rare tumors. However, due to the dismal prognosis, they constitute a significant clinical problem. Standard treatment methods (possibly complete tumor resection, radiotherapy, chemotherapy) are not optimal and it is, therefore, necessary to search for new therapeutic methods that can increase the survival rate. One possibility is the use of immune cells expressing a chimeric antigen receptor (CAR). In the following work, we present the latest results of preclinical and clinical studies evaluating the effectiveness of CAR-expressing cells in HGG therapy. Despite promising results of preclinical studies, most of the patients described in the clinical trials who received the CAR-expressing cell died of cancer. However, the transient efficacy of CAR-expressing cells was also observed, resulting in the reduction of tumor mass and prolongation of survival compared to the median survival time of HGG patients treated in a standard way.

**Abstract:**

High-grade gliomas (HGG) account for approximately 10% of central nervous system (CNS) tumors in children and 25% of CNS tumors in adults. Despite their rare occurrence, HGG are a significant clinical problem. The standard therapeutic procedure in both pediatric and adult patients with HGG is the surgical resection of the tumor combined with chemotherapy and radiotherapy. Despite intensive treatment, the 5-year overall survival in pediatric patients is below 20–30%. This rate is even lower for the most common HGG in adults (glioblastoma), at less than 5%. It is, therefore, essential to search for new therapeutic methods that can extend the survival rate. One of the therapeutic options is the use of immune cells (T lymphocytes/natural killer (NK) cells) expressing a chimeric antigen receptor (CAR). The objective of the following review is to present the latest results of preclinical and clinical studies evaluating the efficacy of CAR-T and CAR-NK cells in HGG therapy.

## 1. Introduction

High-grade gliomas (HGG) account for approximately 10% of central nervous system (CNS) tumors in children and 25% of CNS tumors in adults [1,2]. Despite their rare occurrence, they constitute a significant problem because of the perceived low efficacy of the currently used therapeutic methods. There are certain differences in the treatment of patients with HGG depending on, among other factors, the patient’s age, the degree of the maturity of the nervous tissue, the histological type of the tumor and its location and advancement [2]. The most frequently used therapeutic procedure includes the most thorough tumor resection, radiotherapy and chemotherapy [3]. Despite intensive treatment, the 5-year overall survival (OS) in pediatric patients is below 20–30%. This rate is even lower for the most common HGG in adults (glioblastoma—GBM), at less than 5% [4,5]. In the case of GBM, the median survival time is on average approximately 15 months from diagnosis while, due to the recent change in the pediatric HGG (pHGG) classification, there are discrepancies in the OS of pediatric patients ranging from 10 to 73 months [2,6]. The characteristics of pHGG and HGG occurring in adults are presented in Table 1. 

Due to the location of cancer, some tumors are unresectable, and even in the case of resectable lesions, their complete removal is hardly possible due to the infiltrative nature of the tumor [16]. The use of radiotherapy in pediatric patients, whose nervous system is still maturing, is also limited. In turn, the presence of the blood–brain barrier reduces the concentration of chemotherapeutics penetrating the CNS, and due to the systemic toxicity of drugs, administering them at a bigger dose to achieve a higher concentration of chemotherapeutics in the CNS is not possible [2,6]. It is, therefore, mandatory to search for new therapeutic methods that may be more effective and thus extend the survival rate. One of the therapeutic options is the use of immune cells (T lymphocytes/natural killer—NK cells) expressing a chimeric antigen receptor (CAR). The CAR is composed of four regions, namely, the extracellular antigen-binding domain, the spacer region, the transmembrane domain and the intracellular signaling domain [17]. The structure of CAR, as well as the production process of T lymphocytes expressing CAR (CAR-T), is presented in Figure 1. 

Thanks to the antigen-binding domain, CAR binds to the target antigen, which triggers a signaling cascade, ultimately leading to the activation of the immune cell and its effector functions towards the cell on the surface of which the antigen bound by CAR is located [17]. The objective of the following review is to present the latest results of preclinical and clinical studies evaluating the effectiveness of CAR-T and CAR-expressing NK (CAR-NK) cells in HGG therapy in children and adults.

## 2. Materials and Methods

Our research consisted of searching for articles published before 31 October 2023 which dealt with the use of CAR-T and CAR-NK cells in the treatment of high-grade glioma. In order to narrow down the range of relevant literature, we checked in PubMed such notions as “chimeric antigen receptor glioblastoma”, “CAR-T glioblastoma”, “CAR-NK glioblastoma”, “CAR-T GBM”, “CAR-NK GBM”, “CAR-T HGG”, “pHGG CAR”, “CAR diffuse midline glioma”, “CAR DMG” (DMG—diffuse midline glioma) and “CAR-T pediatric brain tumors”. All articles retrieved in the aforementioned manner were assessed against our inclusion and exclusion criteria. Our analysis included works in English covering the topic of preclinical and clinical research on the use of immunotherapy with T or NK cells expressing CAR in HGG. We excluded studies covering in detail the impact of the immunosuppressive tumor microenvironment (TME) on CAR-expressing cells, for it was not within the scope of our review. Articles meeting our inclusion and exclusion criteria were used for analysis in this work (see Figure 2). 

## 3. CAR-T Cell Therapy in High-Grade Glioma

### 3.1. CAR-T Cell Therapy in Pediatric High-Grade Glioma

#### 3.1.1. GD2

Disialoganglioside (GD2) is a subtype of gangliosides which is composed of two sialic acid residues linked with three monosaccharide units. Other types of gangliosides are found physiologically on many tissues in the body, while GD2 is expressed in normal tissues only in the CNS, sensory peripheral nerves, melanocytes, lymphocytes and mesenchymal stem cells [18]. GD2 has gained interest as a target for cancer treatment because it is overexpressed in many types of cancer, including neuroblastoma, retinoblastoma, melanoma, Ewing sarcoma, glioma, osteosarcoma and small-cell lung cancer [18,19]. The mechanisms of action of GD2 in cancer are unknown, but its expression is associated with, among others, tumor proliferation, migration and invasion [20]. In the case of gliomas, an increased activity of the GD2 synthesis pathway is observed in pHGG with the substitution of lysine 27 to methionine in histone H3 (H3K27M), with a relatively low activity of this pathway in H3-wildtype (WT) pHGG [21]. In 2015, the Food and Drug Administration approved the first anti-GD2 antibody for the treatment of neuroblastoma [22]. Therefore, the use of GD2 as a target antigen in CAR-T therapy seems to be justifiable.

**Preclinical studies**: In the study presented by Mount et al., the possibility of using CAR-T cells targeted against the GD2 antigen present on H3K27M mutated DMG cells was shown. The authors constructed CAR-T cells with the tumor necrosis factor receptor superfamily 9 (4-1BB) co-stimulatory domain and showed that they killed DMG cells in vitro. In in vivo conditions, orthotopic mouse xenografts of diffuse intrinsic pontine glioma (DIPG) were prepared. Anti-GD2 CAR-T cells were administered to mice 7–8 weeks after transplantation. In one of the study cohorts, lethal toxicity was observed in several treated mice. It turned out that the administration of anti-GD2 CAR-T cells is accompanied by extensive inflammation and hydrocephalus after the destruction of the tumor cells. The authors emphasize that the effectiveness and safety of CAR-T cell therapy may depend on the neuroanatomical location of the tumor, and in the case of locations particularly sensitive to edema, this therapy may turn out to be especially dangerous. In the remaining mice, a reduction in tumor burden was observed within 40 days after the administration of CAR-T cells. On Day 50, mouse brains were histologically examined and significant tumor eradication was found in the mice treated with anti-GD2 CAR-T cells. However, a small number of GD2^−^ tumor cells were also observed, which may indicate the possibility of antigen escape [21]. As a solution to this problem, Ramaswamy et al. suggested the simultaneous use of epigenetic drugs, such as histone demethylase inhibitors or histone deacetylase inhibitors, as well as immunotherapy using other cell surface markers [23]. Furukawa et al. highlighted another aspect of the use of GD2 as a potential target for CAR-T cells. It has been shown that peripheral T cells, as a result of T-cell receptor (TCR)-dependent signaling, can express GD2 on their surface [24]. Therefore, a fratricide effect could occur, i.e., CAR-T cells killing each other [25]. This is another issue that requires further research [24].

In another study, de Billy et al. investigated the use of a combination of anti-GD2 CAR-T cell therapy with a kinase inhibitor. The authors screened 42 kinase inhibitors, analyzing their effect on DIPG cells and anti-GD2 CAR-T cells. It turned out that dual insulin-like growth factor 1 receptor/insulin receptor (IGF1R/IR) kinase inhibitors, BMS-754807 (BMS) and linsitinib (LIN), effectively inhibited the proliferation of glioma cells without negatively affecting CAR-T cells. The IGF1R/IR pathway regulates the immune system and, among other things, promotes macrophage polarization from the pro-inflammatory M1 to the pro-tumor M2 phenotype. In in vitro conditions, a greater effectiveness of BMS/anti-GD2 CAR-T and LIN/anti-GD2 CAR-T combination therapy was observed compared to using them individually. The authors also compared the effects of LIN and BMS on anti-GD2 CAR-T. It turned out that the presence of LIN reduces the exhaustion of CAR-T cells and induces a more T-cell-immature phenotype. Therefore, a combination of LIN and anti-GD2 CAR-T cells was used in the in vivo model. It was observed that the LIN/anti-GD2 CAR-T combination required fewer CAR-T cells to achieve effective treatment in comparison with the use of anti-GD2 CAR-T cells alone. Reducing the number of required CAR-T cells while achieving the therapeutic goal is associated with the reduction of potential side effects of CAR-T therapy [26].

**Clinical studies:** The first report on the outcome of anti-GD2 CAR-T treatment in children and young adults with DMG was published in 2022. In this Phase I trial, researchers created anti-GD2 CAR-T cells with a 4-1BB co-stimulatory domain and T-cell receptor T3 zeta chain (CD3ζ) as a signaling domain. Their safety and tolerability were evaluated in four patients, aged 5–25, diagnosed with DMG with H3K27M mutation. All patients received the same dose of the drug intravenously—1 × 10^6^ anti-GD2 CAR-T cells/kg. Patient 1 was a 14-year-old girl with DIPG and with early radiographic signs of tumor progression. Six days after the administration of CAR-T cells, the patient experienced Grade 1 cytokine release syndrome (CRS) with cranial nerve symptoms and tumor inflammation-associated neurotoxicity (TIAN). She was treated with tocilizumab (humanized monoclonal antibody against the interleukin-6 receptor) and corticosteroids. On Day 9, the patient experienced a fever, hypertension, decreased responsiveness, hemiplegia and extensor posturing, which required drainage of 10 mL of cerebrospinal fluid (CSF) and the administration of anakinra (human interleukin 1 receptor antagonist protein). One month after the administration of CAR-T cells, tumor progression was observed, and after 3 months, the patient died. Patient 2 was a 21-year-old man with DIPG, who also showed early signs of clinical and radiographic tumor progression. This patient developed Grade 2 CRS and TIAN 7 days after the infusion of CAR-T cells, and was treated with the use of tocilizumab and anakinra. By the end of the first month, the patient’s neurological condition stabilized, with the exception of residual sixth nerve palsy. This improvement lasted for 2–3 months, after which the neurological symptoms returned. The patient then received a second dose of CAR-T cells—30 × 10^6^ CAR-T cells intracerebroventricularly. Fever and symptoms of neuroinflammation were observed, requiring CSF drainage and treatment with corticosteroids, anakinra and hypertonic saline. Two weeks after the second administration of CAR-T cells, an improvement in the patient’s condition and 27% reduction in tumor volume were observed. The patient received three more intracerebroventricular infusions during further treatment, but died due to an intratumoral hemorrhage 10 months after the first infusion of CAR-T cells. The third patient was a 5-year-old girl with DIPG without any signs of tumor progression. Seven days after the infusion of anti-GD2 CAR-T cells, she experienced Grade 1 CRS and TIAN, and was treated with tocilizumab and anakinra. Within 2 months of the administration of CAR-T cells, a significant improvement in neurological symptoms and an increase in height and weight were observed. Three months after the first infusion, she received 12.9 × 10^6^ CAR-T cells/kg i.c.v. She did not develop any serious symptoms related to the administration of CAR-T cells. The patient died because of tumor progression 7 months after the first administration of anti-GD2 CAR-T cells. The last patient was a 25-year-old woman with an H3K27M^+^ spinal cord DMG with early signs of tumor progression. In this patient, the disease rapidly progressed during the manufacturing of the drug, which disqualified the patient from the study. Therefore, she received 1 × 10^6^ CAR-T cells/kg in an emergency investigational new drug application. On Day 6, she developed Grade 3 CRS, treated with tocilizumab and corticosteroids. Within a month, the tumor volume in the spinal cord decreased by more than 90%, but no improvement in brain metastases was observed. More than 75 days after the administration of CAR-T cells, there was significant tumor progression in the patient’s brain, which required partial tumor resection. The patient was given 50 × 10^6^ GD2-CAR-T cells i.c.v. and within 48 h she developed fever and Grade 3 encephalopathy. She was, therefore, treated with anti-inflammatory drugs. Within 3 weeks, tumor regression was again observed, both in the spinal cord and in some locations in the brain. After the data cut-off, the patient survived 11 months after the first infusion of anti-GD2 CAR-T cells [27].

#### 3.1.2. B7-H3

B7 homologue 3 protein (B7-H3), also known as CD276, is a member of the B7 family of immunomodulatory type 1 transmembrane glycoproteins [28,29]. It occurs in two different isoforms that differ in the structure of the extracellular domain. It may contain one or two pairs of immunoglobulin variable-like and immunoglobulin constant-like domains [30]. The expression of this antigen has been found in many CNS cancers such as GBM, medulloblastoma and DIPG, as well as in other types of cancer, e.g., craniopharyngioma, prostate cancer and squamous cell carcinoma [28,31]. B7-H3 is rarely expressed in healthy tissues [31]. The mechanism of action of B7-H3 is based on co-inhibitory signaling, which, by inhibiting T-cell activation, induces immunosuppression in the TME and tumor growth. B7-H3 has also been reported to be associated with tumor invasion, metastasis and treatment resistance [32].

**Preclinical studies:** One study assessed the in vitro efficacy of anti-B7-H3 CAR-T cells against pediatric tumors of the CNS. B7-H3 expression was shown to be present on all tested samples of DIPG. Moreover, when co-cultured with DIPG patient-derived cells, anti-B7-H3 CAR-T cells intensely produced interferon γ (IFN-γ), tumor necrosis factor α (TNF-α) and interleukin 2 (IL-2) [33]. Haydar et al. profiled the expression of various antigens in 49 patient-derived orthotopic xenograft (PDOX) samples. It turned out that B7-H3 and GD2 are the most consistently expressed, followed by interleukin-13 receptor subunit α-2 (IL-13Rα2 or CD213A2), human epidermal growth factor receptor 2 (HER2) and ephrin type-A receptor 2 (EphA2). The authors decided to focus further research on B7-H3 due to its lack of expression on healthy nerve cells. They created human anti-B7-H3 CAR-T cells and assessed their effectiveness in B7-H3^+^ medulloblastoma and DIPG PDOX models. They showed significant inhibition of tumor progression and increased survival compared to control models. In addition, the authors assessed the efficacy of murine anti-B7-H3 CAR-T cells in an immunocompetent brain tumor model. CAR-T cells showed good efficacy after both local (12 out of 20 mice had complete tumor regression) and intravenous (6 out of 10 mice had complete tumor regression) administration. At the same time, no on-target/off-tumor toxicity was observed [34].

**Clinical studies:** The BrainChild-03 phase I clinical trial is currently underway, assessing the feasibility and safety of B7-H3-specific CAR-T cells in the treatment of children and young adults with DIPG [35]. In 2023, Vitanza et al. presented the results of the first three patients aged 10–22. All patients received the same dose—1 × 10^7^ anti-B7-H3 CAR-T cells per week. In total, they received 10–18 infusions. The most common adverse events occurring in all patients after the administration of the drug were as follows: headache, nausea/vomiting and fever. However, these symptoms resolved within 72 h in each patient, without the need for treatment with corticosteroids or cytokine antagonists. Imaging tests performed 6 months after starting CAR-T cell treatment revealed an increase in tumor mass and the presence of infiltration in two patients. Both of the aforementioned patients were included in the study after initial tumor progression. In the third patient, who showed no signs of progression at enrollment, a reduction in tumor volume by approximately 19% was observed. This patient showed improvement for 12 months after entering the study, but decided not to continue further therapy. The two remaining patients survived 494 and 328 days after the first dose of anti-B7-H3 CAR-T cells [36].

#### 3.1.3. Other Antigens

GD2 and B7-H3 are not the only pHGG target antigens currently under investigation. The literature also contains research results showing the use of HER2 as a target antigen for CAR-T therapy in the treatment of pHGG. HER2 overexpression is found in many cancers such as breast, stomach and ovarian cancer, but also in CNS cancers, including DIPG [37,38,39]. Wang et al. constructed second-generation CAR-T cells targeting HER2^+^ tumor cells. The authors co-cultured anti-HER2 CAR-T cells with DIPG cell lines. CAR-T cells were shown to effectively kill cancer cells with the increased granzyme B release in vitro. In order to evaluate the effector functions of CAR-T cells in vivo, DIPG-implanted mouse models were created. After administering anti-HER2 CAR-T cells to mice, the effects of treatment were observed on a weekly basis. Anti-HER2 CAR-T cells led to a significant reduction in the volume of tumors in the tested mice, and consequently no tumor was detected in these mice 6 weeks after drug administration. Due to the development of graft-versus-host disease (GvHD), the experiment was terminated and it was impossible to assess the potential recurrence caused by the survival of a small number of cancer cells which were undetectable in tests [39]. 

In 2017, Ahmed et al. presented the results of a clinical trial in which 17 patients with GBM, including 7 patients under 18 years of age, were given anti-HER2 CAR-T cells [40]. Due to the introduction of the new 2021 World Health Organization Classification of Tumors of the Central Nervous System, GBM is no longer diagnosed in children [41]. However, due to the fact that this study was based on the previous classification from 2016, researchers used the term “glioblastoma” for childhood cancers [42]. Therefore, we decided to describe the results of this study in pediatric patients in this chapter, while the description of adult patients can be found in the chapter on the treatment of adult patients with GBM. The authors constructed an anti-HER2 CAR expressed by virus-specific T cells (VSTs). The cells created in this way were also co-stimulated via the native TCR. The study included seven children aged 10–17 (median age—14.4), who were administered between 1 × 10^6^ and 1 × 10^8^ anti-HER2 CAR VST cells. One patient experienced Grade 2 seizure and a headache. The disease response was evaluated at 6 weeks after the initial infusion of anti-HER2 CAR VST cells. One patient had partial response, the condition of two patients remained stable and the remaining ones experienced disease progression. The median survival time from the infusion of CAR VST cells was 6.1 months (range 2.7–28.6) [40]. 

Another antigen considered by researchers as a therapeutic target in pHGG is integrin α_v_β_3_ (α_v_β_3_). This integrin consists of two subunits—α_v_ (CD51) and β_3_ (CD61) [43]. It has been reported that α_v_β_3_ is expressed on many cancers, including CNS tumors [43,44,45]. In addition, this integrin is present on newly formed vascular endothelial cells, and it participates in the proliferation and maintenance of the tumor [44,45]. Therefore anti-α_v_β_3_ CAR-T cells would target not only the tumor, but also the surrounding newly formed vessels [45]. Cobb et al. observed that in various DIPG cell lines, α_v_β_3_ expression ranges from moderate to high levels. In order to evaluate the safety of such CARs in humans, α_v_β_3_ expression in normal tissues was also assessed with the use of immunohistochemistry. Zero or minimal expression was observed in most tissues examined, while moderate expression was observed in skin and ovaries. Based on that, the researchers created two anti-α_v_β_3_ CAR-T constructs with cluster of differentiation 28 (CD28) or 4-1BB as co-stimulatory domains. In in vitro conditions, the efficacy of both types of anti-α_v_β_3_ CAR-T cells was assessed in the presence of DIPG cell lines. CAR-T cells were shown to effectively kill tumor cells, and high levels of cytokine secretion such as IFN-γ, IL-2 and TNF-α were found. Then, in vivo CAR-T cell activity was evaluated in mouse models. Anti-α_v_β_3_ CAR-T cells were administered intratumorally to mice 3 weeks after tumor implantation. During the first 2 weeks of treatment, mice treated with anti-α_v_β_3_ CAR-T cells with a CD28 co-stimulatory domain showed rapid tumor regression, while in mice treated with CAR-T cells with a 4-1BB co-stimulatory domain, tumor volume decreased, but it was still detectable. In the case of mice treated with CAR-T cells with a 4-1BB co-stimulatory domain, all of them survived throughout the study (8 weeks), in contrast to mice treated with CAR-T cells with a CD28 co-stimulatory domain, where 39% of them died within the first 11 days. The authors theorized that this may have been due to the toxicity of CAR-T therapy. Ultimately, tumor recurrence was observed in all mice, but compared to the control sample, tumor growth was much slower in the treated mice [45].

### 3.2. CAR-T Cell Therapy in Adult High-Grade Glioma

#### 3.2.1. EphA2

One of the most important classes of tyrosine kinase receptors are Eph receptors. To date, eight related ligands have been described for 14 Eph receptors: 9 EphA receptors and 5 EphB receptors [46,47]. The combination of a ligand present on the cell membrane of one cell with an Eph receptor located on the cell membrane of another cell mainly results in cell repulsion or adhesion. During the embryonic period, Eph receptors and their corresponding ligands regulate angiogenesis and neurogenesis, tissue formation, among other things [48]. The EphA2 receptor is a transmembrane protein with the highest affinity for the Eph A1 ligand. In adults, EphA2 is found mainly in proliferating epithelial tissues. It has been noted that the overexpression of receptors for Eph, particularly Eph2A, contributes to carcinogenesis and tumor progression [49]. It plays an oncogenic role and is an adverse prognostic factor in, among others, breast cancer, lung cancer, melanoma and GBM [50].

**Preclinical studies:** Yi et al. designed three types of EphA2-targeted CAR-T cells: second-generation CAR-T cells with a CD28 co-stimulatory domain, second-generation CAR-T cells with a 4-1BB co-stimulatory domain and third-generation CAR-T cells with both of the aforementioned domains. All three types of CAR-T cells contained a short spacer and single-chain variable fragment (scFv) derived from monoclonal antibody (mAb) 4H5, which recognizes the EphA2 protein epitope in a conformation found only in malignant cells. The authors examined the activity of these CAR-T cells in vitro and in vivo. CAR-T cells, regardless of having an additional co-stimulatory domain, proliferated in the presence of EphA2^+^ tumor cells, exhibited cytolytic activity and produced cytokines: IFN-γ and IL-2. In vivo, over a period of 100 days of observation, all three types of CAR-T cells inhibited tumor growth. However, third-generation CAR-T cells were not characterized by improved anti-GBM activity [51].

In another study, An et al. created two third-generation CAR-T cell types targeting two different EphA2 epitopes, with CD28 and 4-1BB co-stimulatory domains. The CAR-T cells differed in a scFv that was derived from two different mAbs specific for EphA2 epitopes, EphA2-a-CAR-T and the EphA2-b-CAR-T. In vitro, both types of CAR-T cells were activated and then they proliferated in the presence of tumor cells. EphA2-a-CAR-T cells showed higher tumor repression (with E:T ratios of 1:1 and 10:1, 76.05% and over 85%, respectively). In the culture of EphA2-b-CAR-T cells together with glioma cells, high levels of IFN-γ were observed. In vivo both types of CAR-T cells inhibited tumor growth, but EphA2-a-CAR-T cells showed better activity against GBM. The difference in gene expression between the two types of EphA2-CAR-T cells was examined using ribonucleic acid (RNA) sequencing to find out why EphA2-a-CAR-T cells showed higher anti-tumor activity. It turned out that EphA2-a-CAR-T cells expressed receptors for interleukin 8 (IL-8), namely, C-X-C Motif Chemokine Receptor 1 and C-X-C Motif Chemokine Receptor 2. Glioma cells can produce IL-8, so CAR-T cells carrying the aforementioned receptors showed better anti-tumor activity. EphA2-b-CAR-T cells expressed higher levels of IL-8 (i.e., CXCL8) and IFN-γ, which was associated with a weaker response against GBM. IFN-γ secretion induces programmed death-ligand 1 (PD-L1) expression in tumor cells, which promotes T-cell suppression. To evaluate whether the programmed death receptor 1 (PD-1) knockout would enhance the antitumor activity of CAR-T cells in vivo, mice were given anti-PD-1 antibody and EphA2-CAR-T cells. Regardless of the presence of PD-1 antibody, tumor growth in mice treated with EphA2-a-CAR-T cells was significantly inhibited. However, in the case of EphA2-b-CAR-T cells, treatment with PD-1 antibodies increased their antitumor activity, suggesting that PD-1 and PD-L1 interaction is responsible for their inhibition [52].

Muhhamad et al. designed third-generation tandem CAR-T cells (TanCAR-T) specific for IL-13Rα2 and EphA2 and tested their efficacy in vitro and in vivo. In the in vitro assay, TanCAR-T cells showed cytotoxic activity against GBM cells bearing a single (EphA2^+^ or IL-13Rα2^+^) as well as dual (EphA2^+^ and IL-13Rα2^+^) target antigen. Since IL-13 can bind to both IL-13Rα2 and IL-13Rα1, which is physiologically present on normal cells, the authors used the IL-13 mutant when constructing the CAR, which has a low affinity for IL-13Rα1. Thanks to this, normal cells expressing physiological IL-13Rα1 were not killed. In vivo, the antitumor activity of TanCAR-T and CAR-T cells targeting single antigens was compared. All analyzed CAR-T cells inhibited tumor growth, but TanCAR-T cells showed the best antitumor effect [53].

**Clinical studies:** The first outcome of anti-EphA2 treatment in patients with GBM dates back to 2021. Three patients with recurrent GBM were intravenously infused with second-generation CAR-T cells with the co-stimulatory domain 4-1BB targeting EphA2 at a dose of 1 × 10⁶ cells/kg. In the first patient, the primary glioma site was located in the left frontal lobe. Four months after standard treatment, there was a recurrence (in the frontal lobe and corpus callosum). Four days after the intravenous administration of CAR-T cells, there was a seven-day high temperature and on the seventh day the patient also had hypotension. Laboratory tests showed elevated levels of cytokines, i.e., IFN-γ, interleukin 6 (IL-6) and TNF-α. On Day 10 of treatment, edema of the middle and lower lobe of the right lung was found on X-ray (Grade 2 CRS). Dexamethasone included in the treatment resulted in fever resolution on Day 11, normalization of cytokine levels within 3 weeks and a complete disappearance of pulmonary edema on Day 30. After 8 weeks, magnetic resonance imaging (MRI) showed contrasting changes in the corpus callosum. After 12 weeks, further progression of the tumor was observed after intravenous infusion. The patient’s OS time was 181 days. In the second patient, the initial glioma site was located in the left frontotemporal region. Due to tumor recurrence, the patient underwent craniectomy three times. Due to residual EphA2^+^ GBM, he received an infusion of CAR-T cells. No side effects or an increase in serum cytokine levels were observed. An MRI scan performed 4 weeks after the infusion showed a reduction in two lesions in the frontal and parietal lobes. After 12 weeks, there was progression of the frontal lobe lesion and a stable image of the parietal lesion. The OS time was 164 days. In the third patient, the primary site was located in the right frontotemporal region. The patient, 18 months after the initial diagnosis, received CAR-T cells directed against EphA2. Seven hours after the infusion, there was a high fever lasting 6 days and an increase in cytokine levels, especially IFN-γ. On Day 6 after the infusion, pulmonary edema occurred, which resolved after 15 days of treatment with dexamethasone. Grade 2 CRS was noted. On Day 7, body temperature normalized, and cytokine levels normalized within 3 days. After 4 weeks, a follow-up MRI scan showed tumor progression, which did not regress after another 4 weeks. Survival time was 86 days. The occurrence of pulmonary edema in two of the three patients may have been related to the effect of CAR-T cells on the EphA2 receptor physiologically found in the lung tissue [54]. It was observed that the intravenous administration of CAR-T cells causes their migration into the lung tissue [55].

#### 3.2.2. IL-13Rα2

IL-13Rα2 is an extramembrane protein that includes a signaling sequence and a short intramembrane domain [56]. Under physiological conditions, IL-13 binds to the IL-13Rα1 receptor, leading to the activation of downstream Janus kinases—signal transducers and activators of transcription protein signaling—which results in apoptosis and the cessation of further cell proliferation. Compared to normal brain cells, cancer cells in over 75% of patients diagnosed with GBM express the IL-13Rα2 decoy receptor, for which IL-13 has high affinity [57,58]. In this case, signaling is not activated, which results in further proliferation of cancer cells. In somatic tissues, IL-13Rα2 expression is found only in the testis. Anti-IL-13Rα2 CAR-T cells may be an effective therapeutic strategy in GBM patients because the normal brain tissue does not express IL-13Rα2, thus limiting the off-target effects of CAR-T cells [59]. In other tumors, i.e., pancreatic cancers, breast carcinoma and ovarian cancer, the described expression was observed [60]. These reports have encouraged the development of antigen-specific therapies targeting IL-13Rα2.

**Preclinical studies:** Xu et al. created third-generation CAR-T cells with two co-stimulatory domains, CD28 and 4-1BB, directed against IL-13Rα2, and tested their efficacy in vitro and in vivo. Since it was noted that the use of mouse-antibody-based CAR-T cells induces an immune response and increases the risk of recurrence, the authors created two types of CAR-T cells. The first one contained mouse-derived scFv (m-IL-13Rα2), and the second one contained humanized scFv (h-IL-13Rα2) with a CD28 transmembrane domain. In vitro, both types of CAR-T cells inhibited tumor cell growth, with m-IL-13Rα2 CAR-T cells releasing higher levels of IFN-γ and IL-6 compared to h-IL-13Rα2 CAR-T cells. In in vivo models, both types of IL-13Rα2 CAR-T cells inhibited tumor growth and prolonged the survival of mice [61].

To test the effect of the presence of co-stimulatory domains in CAR constructs on the ability to selectively kill IL-13Rα2^+^ glioma cells, the researchers compared four types of CARs. Starr et al. engineered IL-13 mutein-containing first-generation CAR-T cells, second-generation CAR-T cells with a CD28 or 4-1BB co-stimulatory domain and third-generation CAR-T cells with CD28 and 4-1BB co-stimulatory domains. In vitro, of all four CAR-T cell types, second-generation CAR-T cells with the 4-1BB co-stimulatory domain showed the highest durability and anti-tumor efficacy. In in vivo conditions, mice were injected intratumorally with 1 × 10⁵ anti-IL-13Rα2 CAR-T cells, and all of the four CAR-T cells variants showed anti-tumor activity. However, it was the second-generation CAR-T cells with the 4-1BB co-stimulatory domain that contributed the most to the reduction in GBM size. More than 60% of individuals who received the aforementioned CAR-T cell type achieved long-term tumor-free survival after 152 days. In addition, compared to the aforementioned CAR-T cells, second-generation CAR-T cells with a CD28 co-stimulatory domain showed cytolytic activity against cells that physiologically possess IL-13Rα2 antigen in vitro and in vivo [62].

Newman et al. investigated the impact of the co-expression of IL-13Rα2 and epidermal growth factor receptor variant III (EGFRvIII) in GBM progression. The results of the study showed that overexpression of IL-13Rα2 alone causes only tumor invasion, without further proliferation. When two oncoproteins (EGFRvIII and IL-13Rα2) are expressed, an increased EGFRvIII tyrosine kinase activity is observed, which results in tumor growth. The survival rate of patients with EGFRvIII^+^ glioma and high IL-13Rα2 expression was significantly lower than that of patients with low IL-13Rα2 expression. High levels of ligands for IL-13Rα2, i.e., IL-13 or chitinase-3-like protein 1, also had a negative impact on the survival of patients. In the in vivo study, tumor volume in the mice implanted with EGFRvIII^+^ and IL-13Rα2^+^ glioma cells increased significantly in comparison with the mice implanted with WT epidermal growth factor receptor (EGFR) glioma cells. The former had a significantly shorter progression-free time (median survival of 40.5 vs. 90 days, respectively). The results of the study suggest that targeting both receptors should be considered in patients with GBM [63]. This kind of strategy was used in the study of Schmidts et al. They created dual-specific, second-generation, 4-1BB co-stimulatory domain TanCAR-T cells targeting EGFRvIII and IL-13Rα2 simultaneously. The researchers generated two GBM cell lines with EGFRvIII or IL-13Rα2 expression (i.e., EGFRvIII^+^/IL-13Rα2¯ and EGFRvIII¯/IL-13Rα2^+^). The in vitro results confirmed that TanCAR-T cells showed superior efficacy against tumor cells containing only the sole target antigen compared to monospecific CAR-T cells (i.e., anti-EGFRvIII CAR-T cells and anti-IL-13Rα2 CAR-T cells). TanCAR-T cells showed more rapid and complete cytotoxicity than anti-EGFRvIII CAR-T cells and anti-IL-13Rα2 CAR-T cells. After 60 h of exposure, TanCAR-T showed approximately 20% higher levels of cytotoxicity than anti-EGFRvIII CAR-T cells and approximately 40% higher levels of cytotoxicity than anti-IL-13Rα2 CAR-T cells. In the in vivo study, the population of mice implanted with mixed GBM was divided into four groups: those receiving control T cells, anti-EGFRvIII CAR-T cells, anti-IL-13Rα2 CAR-T cells or TanCAR-T cells. An incomplete response was observed in mice treated with monospecific CAR-T cells, and survival times were comparable to the control group. Mixed tumors maintained heterogeneous antigen expression after their implantation into mice, but the administration of monospecific CAR-T cells resulted in a loss of expression of the target antigen. Meanwhile, all mice treated with TanCAR-T cells achieved a complete and durable response (survival rate of more than 150 days) [64].

Hegde et al. created second-generation TanCAR-T cells containing an HER2-specific scFv fragment, an IL-13 mutein that binds to IL-13Rα2 as well as the CD28 co-stimulatory domain. In vitro, TanCAR-T cells showed cytolytic activity against all tested tumor cells (HER2^+^, IL-13Rα2^+^, HER2^+^ and IL-13Rα2^+^). In vivo, mice treated with CAR-T cells targeting a single antigen achieved only transient, but at the same time significant, tumor regression (median progression-free survival, PFS, was 14 days). Median PFS in individuals treated with TanCAR-T cells was 36 days (range 32–110 days). The median OS time in mice treated with anti-HER2 CAR-T cells, anti-IL-13Rα2 CAR-T cells and TanCAR-T cells was 53 days, 55 days and 86 days, respectively. In the case of glioma recurrence, the tumor cells lost the antigens that were targeted by TanCAR-T cells. Additionally, the researchers compared the efficacy of TanCAR-T cells with bispecific T cells (biCAR-T cells) in in vivo models. The survival time of mice treated with biCAR-T cells was significantly shorter than that of the mice treated with TanCAR-T cells, at 85 days (range 57–129 days) and more than 140 days, respectively. Three out of ten mice (30%) treated with biCAR-T cells and all ten mice treated with TanCAR-T cells survived until the end of the experiment [65].

In another study conducted by Bielamowicz et al., the authors designed second-generation CAR-T cells, with a CD28 co-stimulatory domain, targeting three antigens: HER2, IL-13Rα2 and EphA2. The trivalent CAR-T cells were named UCAR-T cells and their anti-tumor activity was tested. In vitro, UCAR-T cells showed anti-tumor activity against glioma cells positive for one of the aforementioned antigens. Nearly one hundred percent of tumor cells with the expression of all three antigens in the presence of UCAR-T cells underwent cytolysis. In vivo, a transient anti-tumor response was observed in mice that received a dose of 3 × 10⁶ CAR-T cells targeting one (IL-13Rα2) or two (IL-13Rα2 and Eph2A) antigens. In individuals that received UCAR-T cells, a more durable anti-tumor response was maintained and they lived more than 60 days. This study proves that antigen-specific therapy targeting more than one target point allows for the compensation of antigen escape and enhancement of effector T-cell function [66].

**Clinical studies:** The first outcome of the use of anti-IL-13Rα2 CAR-T cells in patients with GBM dates back to 2015. Three patients with recurrent GBM received up to twelve intracranial infusions at a maximum cumulative doses of 10.6 × 10⁸ CAR-T cells. A transient antiglioma response was observed in two out of three patients. Adverse events were observed, including temporary neurological deficits and headaches that disappeared after treatment. Moreover, in the analyzed brain tissue in one of the patients, the general expression of IL-13Rα2 decreased in comparison with the baseline. Since one of the conditions for including patients in the clinical trial was relapse after previous treatment, the authors reported in the results of the study the total survival time of patients from relapse, which was 10.3 months, 8.6 months and 13.9 months, respectively [59].

Another study presented a clinical case of a 50-year-old patient diagnosed with multifocal GBM and treated with the use of anti-IL-13Rα2 CAR-T cells. The initial tumor was located in the right temporal lobe and the patient received standard treatment. Six months after diagnosis, there was a recurrence of the disease in the meninges and both hemispheres of the brain. Before CAR-T cell therapy was administered, the patient underwent surgical resection of three of the five glioma foci. The patient received multiple infusions of CAR-T cells. The first and subsequent doses were 2 × 10⁶ and 10 × 10⁶ CAR-T cells, respectively, and were administered into the surgical locus and later into the ventricular system. Initial infusions resulted only in a lack of tumor progression of the temporo-occipital region. An MRI showed new metastatic lesions in the spine, the progression of unresectable tumor and two new focal areas near the previously removed lesions. Subsequently, ten intratumoral infusions were administered to allow CAR-T cells to move into the foci. A radical reduction in the size of all tumor foci was observed after three infusions, and after the fifth administration, tumors had shrunk by 77–100%. During a consolidation phase involving the administration of five additional intratumoral infusions, all disease foci resolved. The patient resumed activity, and the clinical response persisted for 7.5 months after starting anti-IL-13Rα2 CAR-T cell therapy. After 7.6 months of therapy, the disease recurred in four new locations that were not adjacent to the previous tumors. Intraventricular administration of CAR-T cells therapy showed greater ability to inhibit tumor growth at distant sites than the intracavitary administration of CAR-T cells. Both intraventricular and intracavitary forms showed a low toxicity profile. Reported side effects in the form of a headache, generalized fatigue, muscle pain and olfactory disturbances were observed within 72 h after the infusion of CAR-T cells [67].

#### 3.2.3. EGFRvIII

The EGFR is a transmembrane protein that is part of the human epidermal growth factor receptor family. It takes part in cell migration, adhesion, differentiation and apoptosis [68,69]. Under pathological conditions, there is constant ligand-independent activation that promotes the growth of abnormal cells [70]. Changes in the EGFR gene, i.e., deletion of exons 2 to 7 and the formation of a glycine residue at the junction of exon 1, result in the expression of EGFRvIII. Due to the loss of 267 amino acids in the extracellular domain, the receptor loses its ability to bind physiological ligands and is still active. About 25% to 81% of the patients with newly diagnosed GBM express EGFRvIII. It has been noted that low-grade gliomas rarely have superficial EGFRvIII [71]. In GBM cells, EGFRvIII co-expresses with an unaltered form of the EGFR receptor (WT EGFR). The consequence of introducing EGFRvIII into cells that initially express WT EGFR is increased proliferation, angiogenesis and tumor invasion [72]. Therefore, EGFRvIII appears to be a reasonable target for the therapy with the use of CAR-T cells.

**Preclinical studies:** Epitope 806 is present in the extracellular domain of EGFR. It is masked in the normal conformation of the EGFR protein. As a result of truncation of the EGFR extracellular domain, which is present in EGFRvIII or EGFR overexpression, e.g., WT EGFR, epitope 806 becomes an available target for anticancer therapies [73]. Ravanpay et al. created second-generation CAR-T cells possessing an extracellular binding domain derived from an mAb against epitope 806 (mAb806), a 4-1BB co-stimulatory domain and truncated EGFR (EGFR806-CAR-T cells). The EGFR fragment had no mAb806 binding site, and was only a transduction marker and ablation target. Three EGFR806-CAR-T cells variants differing in the length of the extracellular spacer domains (short, medium, long) were produced. The authors examined that EGFR806-CAR-T cells with short spacers in the extracellular domain induced the strongest production of effector cytokines IL-2, IFN-γ and TNF-α after tumor recognition. The in vitro study was conducted on three human glioma cell lines. In EGFR^+^ cell lines, EGFRvIII expression was not detected by RNA sequencing, confirming that EGFR806-CAR-T-cell activity is not dependent on the presence of EGFRvIII. EGFR806-CAR-T cells induced effective lysis of all tumor cell lines. In an in vivo study, mice received a single intratumoral dose of EGFR806-CAR-T cells 7 days after the injection of human glioma cells. Tumor regression occurred in all of these individuals. Half of the mice treated with EGFR806-CAR-T cells lived to the established end date of the study (90 days) [74].

Chen et al. designed second-generation mouse CAR-T cells targeting EGFRvIII with a CD28 co-stimulatory domain and extracellular scFv derived from antibody 806 (806-28Z-CAR-T cells) and tested their ability against mouse glioma cells in vitro and in vivo. Mouse EGFRvIII^+^ GBM cell lines were modified by inserting a human EGFR806 epitope. Anti-EGFRvIII CAR-T cells showed dose-dependent cytolytic activity against EGFRvIII^+^ glioma cells. Only when anti-EGFRvIII CAR-T cells were incubated with the cell lines of GBM significant levels of cytokines such as IL-2, TNF-α and IFN-γ were detected. High levels of granzyme B and serine protease (cytotoxic T-lymphocyte granules) were also detected in these samples. In vivo, anti-EGFRvIII CAR-T cells, which were intravenously administered, resulted in a significant tumor growth inhibition and prolonged mouse survival. Increasing the dose of anti-EGFRvIII CAR-T cells resulted in a complete regression of GBM. The concentration of granzyme B in mouse serum correlated with the dose of CAR-T cells administered. Then, the cured mice were re-injected with EGFRvIII^+^ or EGFRvIII^−^ GBM cells. On the third day of the study, no increase in EGFRvIII^+^ tumor volume was observed at all in the previously treated mice. In contrast, an increase in EGFRvIII¯ tumor volume was noted in both treated and untreated mice; however, the growth rate was slower in the previously treated mice. On Day 13, a threefold increase in tumor volume was observed in the control sample, while a twofold decrease in GBM volume was observed in the previously treated mice. These results suggest that 806-28Z-CAR-T cells had the ability to activate anti-tumor memory in mice [75].

In another study, Hua et al. constructed second-generation CAR-T cells with a CD28 co-stimulatory domain and third-generation CAR-T cells with CD28 and 4-1BB co-stimulatory domains targeting both EGFR and EGFRvIII. The scFv was derived from humanized antibody M27 (scFv M27) or from mAb 806 recognizing the EGFR epitope and EGFRvIII. In vitro third-generation CAR-T cells with scFv M27(CAR-T M27-28BBZ) showed the highest lysis capacity of glioma cells and showed no significant cytotoxicity to healthy tissues. The aforementioned CAR-T cells in the presence of EGFR or EGFRvIII secreted high levels of cytokines, i.e., IL-2, TNFα and IFN-γ. In vivo, mice were injected intravenously with CAR-T M27-28BBZ cells on Days 12 and 15 of the study. On Day 42, a reduction in the volume of EGFRvIII-expressing gliomas was observed. Eight days later, tumor regression was also observed in mice implanted with EGFR-expressing gliomas. Next, the authors tested the ability of CAR-T M27-28BBZ cells to inhibit the growth of glioma cells expressing EGFRvIII, EGFR or both antigens in orthotopic GBM xenografts models. On Day 6, mice were injected intravenously with CAR-T M27-28BBZ cells. A total of 3–4 weeks later, a significant tumor growth inhibition was observed in individuals treated with the aforementioned cells. Interestingly, survival time for several mice treated with CAR-T M27-28BBZ was longer than 80 days [76].

Anti-EGFRvIII CAR-T cells were also the subject of a study by Abott et al. who created second-generation CAR-T cells with an scFv showing high affinity for EGFRvIII and with a CD28 co-stimulatory domain and tested their ability to kill glioma cells in vitro and in vivo. In vitro, engineered CAR-T cells showed antitumor activity against EGFRvIII glioma cell lines. In vivo, after one week, mice that received a single peripheral infusion of anti-EGFRvIII CAR-T cells showed a reduction in tumor size. After two weeks, the size of tumor in the treated mice was below the detection limit. The situation was stable for 2 weeks of observation. Meanwhile, in the control group, tumors reached a large size. The results of this study confirmed the high specificity of engineered CAR-T cells targeting the EGFRvIII antigen [77].

**Clinical studies:** The first outcome of anti-EGFRvIII treatment in patients with GBM dates back to 2017. Ten patients, aged 45–76 (median 59.5), with recurrent GBM were enrolled in the human clinical trial involving the intravenous administration of a single dose of anti-EGFRvIII CAR-T cells. Nine out of ten patients had multifocal disease. The tenth patient originally had partially unresectable GBM located in the thalamus and midbrain. Despite the prior administration of temozolomide (alkylating agent), which can cause side effects such as leukopenia/neutropenia, all patients were able to produce adequate numbers of anti-EGFRvIII CAR-T cells (target dose of 1 × 10⁸ to 5 × 10⁸ anti-EGFRvIII CAR-T cells). Clinically significant neurological events were observed in three patients, which were managed by the administration of siltuximab (chimeric monoclonal antibody against the interleukin-6) (in the first patient) and high doses of corticosteroids and siltuximab (in the second patient), including the clinical observation of a secondary hemorrhage, which did not require surgery (in the third patient). A 59-year-old female patient achieved an 18-month remission. The median OS of the patients was 251 days (approximately 8 months). The majority of patients underwent neurosurgical intervention following therapy; therefore, progression-free survival rates could not be assessed [78]. In May 2021, a case report of the aforementioned 59-year-old patient was published. She survived 36 months after relapse and 34 months after a single infusion of 9.2 × 10⁷ anti-EGFRvIII CAR-T cells. On Day 7, the patient developed mild flu-like symptoms, i.e., arthralgia, myalgia and headaches. Three months later, the patient was treated with dexamethasone for increasing headaches. After 104 days, the patient underwent surgical resection of the tumor. The results of the immunohistochemical examination showed a reduction in superficial EGFRvIII expression compared to baseline (from 78% to 3.7%). The patient did not require additional chemotherapy for 18 months after the second surgery. A follow-up MRI scan 15 months after surgical treatment showed gradual progression of the tumor, and 34 months after the administration of CAR-T cells, the patient died of infectious osteomyelitis [79].

The results of another clinical study were presented by Goff et al. Eighteen patients were qualified for the peripheral infusion of third-generation anti-EGFRvIII CAR-T cells containing the co-stimulatory domains of CD28 and 4-1BB at a dose of 6.3 × 10⁶ to 2.6 × 10¹⁰. Thirty-three patients were included in the clinical screening study, but ultimately only eighteen patients met the eligibility requirements for the clinical trial. Eligible patients had EGFRvIII^+^ GBM and were found to have radiological recurrence after radiotherapy, chemotherapy or surgical resection. The median interval between diagnosis and the administration of peripheral anti-EGFRvIII CAR-T cells was 11.1 months. Prior to the peripheral infusion of CAR-T cells, patients received cyclophosphamide (antineoplastic agent metabolized to active alkylating metabolites) for two days, followed by fludarabine (purine analogue) for five days. A total of 12 h after the administration of CAR-T cells, patients received intravenously a low dose of IL-2. One patient developed acute dyspnea and desaturation approximately one hour after the administration of anti-EGFRvIII CAR-T cells, and intubation was required. Unfortunately, severe hypotension occurred, leading to the patient’s death within four hours. Another patient developed dyspnea four hours after the administration of the drug and required ventilation with continuous positive airway pressure. After a short stay in the intensive care unit, the patient was dismissed home. Ten patients required additional brain imaging due to the fact that they had Grade 2 neurological symptoms or suspected seizures. These symptoms were treated with corticosteroids and/or antiepileptic drugs. At follow-up MRI examination, there was no objective response to the treatment. The first follow-up examination showed progressive disease with a median PFS of 1.3 months. A total of 16 out of 17 patients had disease progression less than three months after the infusion of CAR-T cells. Three patients required immediate treatment with bevacizumab (humanized monoclonal antibody against VEGF-A (vascular endothelial growth factor A)), one patient required surgical resection and seven patients were provided with palliative care. The median survival time of all patients was 6.9 months. One patient received no further treatment after anti-EGFRvIII CAR-T cell therapy and was still alive at 59 months. Two other patients survived 13.1 and 13.6 months, respectively. The described treatment neither produced objective tumor regression nor appeared to delay progression or prolong survival in patients with recurrent GBM [80].

In a study reported by Tang et al., ten patients with recurrent or progressive surface-expressing EGFRvIII GBM were enrolled in the human clinical trial. The patients were administered a dose of 1–5 × 10⁸ second-generation anti-EGFRvIII CAR-T cells with a 4-1BB co-stimulatory domain in a single intravenous infusion. One patient was excluded from the statistical analysis due to the PFS time of 615 days. Seven out of ten patients underwent surgical resection after CAR-T cell therapy. The median PFS was 80 days [81].

#### 3.2.4. Other Antigens

Researchers have considered designing CAR-T cells targeting other antigens than those mentioned above, namely, GD2, B7-H3, prominin-1 (CD133), cell surface glucose-regulated protein 78 (csGRP78) and HER2.

Prapa et al. have tested the activity of anti-GD2 CAR-T cells against GBM. In vitro, autologous anti-GD2 CAR-T cells exhibited great anti-tumor activity. In vivo, the survival rate of mice that received a total intravenous dose of 3.4 × 10⁶ anti-GD2 CAR-T cells was comparable to the control group. The intracerebral administration of anti-GD2 CAR-T cells along with glioma cells delayed tumor growth by 2 weeks or 3–4 weeks, depending on the dose (2 × 10⁵ or 5 × 10⁵ anti-GD2 CAR-T cells, respectively). Regardless of the route of the administration of the therapy, side effects were not observed [82]. In another study, researchers tested the efficacy of third-generation anti-GD2 CAR-T cells. In vivo, mice received 1.5 × 10⁶ CAR-T cells intravenously from a healthy donor, while the second group received CAR-T cells from a patient diagnosed with GBM. The median survival time of the mice was 52 and 53 days, respectively. Despite the observed lack of neurotoxicity and a slowdown in tumor growth, the therapy did not show a curative effect. The researchers, in order to improve the efficacy of the treatment, increased the dose range from 10⁶ to 10⁷ cells and modified anti-GD2 CAR-T cells with the addition of interleukin 15 (IL-15). The average survival time of the mice treated with CAR-T cells with incorporated IL-15 was 63.5 days, 16.5 days longer than the average survival time of the mice treated with anti-GD2 CAR-T cells. In addition, 4 weeks after the beginning of treatment, the mice treated with IL-15-containing CAR-T cells had prolonged survival and a complete response rate was at a level of 50% [83].

In order to reduce the side effects and cytotoxicity outside the tumor which are associated with the administration of CAR-T cells, Saleh et al. developed a Reversed CAR (RevCAR) system consisting of RevCAR-T cells that have a peptide epitope in the extracellular domain and a bispecific target module (RevTM). In this case, RevTM is involved in the recognition of the GD2 or EGFR antigen located on glioma cells and the target epitope of RevCAR-T cells. Only after both molecules are recognized do RevCAR-T cells become activated. In vitro, only RevCAR-T cells in the presence of specific and appropriately matched RevTMs showed cytolytic activity. Subsequently, dual-RevCAR-T cells were created to target both antigens (GD2 and EGFR). In vitro, GBM cell lines underwent cytolysis by these CAR-T cells only in the presence of two specific RevTMs. In the in vivo study, mice were divided into five groups and injected with glioma cells only, glioma cells and dual-RevCAR-T cells or glioma cells and dual-RevCAR-T cells with one of the two or both RevTMs, respectively. On Day 9 of the study, only in animals that received dual-RevCAR-T cells with both RevTMs, tumor growth was markedly inhibited [84].

Glucose-regulated protein 78 (GRP78) is involved in the process of protein folding and assembly in the endoplasmic reticulum of normal cells. During neoplasia, the level of GRP78 increases, which is partially incorporated into the cell surface (csGRP78). The presence of this expression has been shown to correlate with glioma malignancy, tumor invasiveness and poorer response to treatment [85]. Therefore, Wang et al. constructed second-generation CAR-T cells with a co-stimulatory domain of 4-1BB and a Pep42 peptide that binds the csGRP78 (Pep42-BBZ-CAR-T cells), and tested their anti-tumor efficacy in vitro and in vivo. In vitro, these CAR-T cells released IFN-γ and showed anti-tumor activity against GBM lines and glioma stem cells expressing csGRP78. In vivo, mice with severe immunodeficiency were subcutaneously implanted with the GBM tumor xenograft model. Mice were intravenously injected with 1 × 10⁷ Pep42-BBZ-CAR-T cells, which caused a slowdown in tumor growth. In addition, the expansion of the aforementioned cells into other organs was insignificant, so they were characterized by safety and specificity of activity [86].

CD133 is a glycoprotein recognized as a marker of cancer stem cell growth. Its expression is associated with resistance to standard treatment, poor prognosis and an increased risk of GBM recurrence [87]. The CD133 surface antigen, along with the cluster of differentiation 34 antigen, appear on the hematopoietic cells of the bone marrow [88]. The authors engineered second-generation CAR-T cells with a CD28 co-stimulatory domain, carrying an scFV containing a fragment of an antibody targeting CD133 (anti-CD133 CAR-T cells). In vitro, in the presence of GBM CD133^+^ cell lines, CAR-T cells showed a dose-dependent anti-tumor response. In vivo, mice intracranially received two doses of anti-CD133 CAR-T cells, which contributed to significant tumor regression and prolonged OS time. After CAR-T therapy, residual tumors were analyzed by flow cytometry, the result of which confirmed the high efficacy of the applied therapy against CD133^+^ GBM. Anti-CD133 CAR-T cell therapy does not significantly reduce the number of human hematopoietic stem and progenitor cells and has no significant effect on hematopoiesis [89].

**Clinical studies:** In 2023, Liu et al. published the results of a study in which four patients diagnosed with GD2 antigen-positive GBM were administered GD2-specific fourth-generation CAR (4SCAR)-T in intravenous and intrathecal infusions at a dose of 2.5 × 10⁶/ kg and 1.0 × 10⁵/kg, respectively. All patients underwent primary surgical resection. Each patient received GD2-specific 4SCAR-T intravenously, and in the case of two patients, intracavitary administration of GD2-specific 4SCAR-T was also performed. A 63-year-old female patient with intravenous and intracavitary drug administration had Grade 2 epileptic seizure and Grade 3 headache. GD2-specific 4SCAR-T cells in the patients’ peripheral blood were assessed by reverse transcription quantitative polymerase chain reaction every week for 4 weeks. The cells proliferated for 1–3 weeks, after which their frequency remained low. Two patients achieved partial response without further therapy as a result of a single infusion, with partial regression of the tumor location on an MRI. In addition, in one of these two patients, partial response persisted for 24 months. In other patients, the disease progressed after the infusion of 4SCAR-T cells, and the patients survived for 6 and 23 months, respectively [90].

The case report of a 56-year-old patient diagnosed with recurrent GBM who was given second-generation CAR-T cells therapy with a 4-1BB co-stimulatory domain targeting B7-H3 dates back to 2021. The primary site of GBM was located in the left frontal and parietal lobes. The patient underwent craniotomy twice within two years and received standard treatment. The recurrent site of GBM showed the heterogeneous expression of B7-H3 antigen. The patient received seven cycles of intracavitary therapy with appropriate doses of anti-B7-H3 CAR-T cells: the first cycle 4 × 10⁸, the next four cycles 1 × 10⁷, the sixth cycle 1.5 × 10⁷ and the seventh cycle 2 × 10⁷. During the first five cycles of treatment, the patient reported unresponsive headaches, so the dose of CAR-T cells was not increased. After the third cycle of treatment, significant tumor regression was observed on an MRI (Day 49 after CAR-T cell therapy began). The response persisted for more than 7 weeks. Disorders of consciousness and lethargy occurred after the sixth and seventh infusions of CAR-T cells. On Day 87 of therapy, the recurrence of the tumor was visible on an MRI. The patient resigned from taking part in the clinical trial after seven cycles of treatment [91].

In a study presented by Ahmed et al., 17 patients with GBM were treated with the use of anti-HER2 CAR VST cells. In the case of 10 adult patients without prior lymphodepletion whose median age was 60 years (range 30–69 years), at least one infusion of 1 × 10⁶ to 1 × 10⁸ anti-HER2 CAR VST cells was performed. In a follow-up MRI scan performed 6 weeks after the administration of anti-HER2 CAR VST cells, half of the patients achieved stable disease, while the other half had progression of GBM. The median survival time from the first administration of anti-HER2 CAR VST cells was 9.4 months (range 2.4 to 28.4). Two patients were alive with stable disease through 28.8 and 24.4 months of follow-up. One patient experienced adverse events in the form of epileptic seizure and headache associated with the infusion of anti-HER2 CAR VST cells [40]. A description of the results of patients under 18 years of age has already been presented in the Section 3.1.3 devoted to pediatric patients.

### 3.3. Recruiting Clinical Trials

The effectiveness of CAR-T cells in preclinical studies translated into the initiation of clinical trials. Currently, 21 recruiting Phase 1 clinical trials are being conducted, testing the possibility, efficacy and safety of using CAR-T cells targeting many antigens such as GD2, B7-H3, IL-13Rα2, matrix metalloproteinase-2 (MMP-2), HER2, natural killer group 2 member D protein (NKG2D) and cluster of differentiation 70 in pHGG and adult HGG therapy. More information about currently recruiting clinical trials is presented in Table 2.

## 4. CAR-NK Cell Therapy in High-Grade Glioma

Despite the unquestionable success of CAR-T cells in the treatment of some hematological malignancies, as well as intensive research to improve their effectiveness in solid tumors, attention should be paid to certain limitations of CAR-T cells, which prompted researchers to start working on CAR-NK cells. In the case of CAR-T cells, due to the risk of developing GvHD, autologous preparations are currently used, which means that if patients fail the leukapheresis process due to lymphopenia, they will be excluded from therapy. Research is ongoing on allogeneic CAR-T cell preparations, which, thanks to additional genetic modifications, would maintain the desired safety profile, but this will further extend the production time of CAR-T cells and increase the cost of preparation. Studies have shown that allogeneic CAR-NK cells do not cause GvHD, which opens the possibility of producing ready-made CAR-NK cell preparations available to patients off the shelf. This will contribute to increasing the availability of therapy by reducing the costs and waiting time for the preparation [111,112,113]. Additionally, the administration of allogeneic CAR-NK cells may increase the efficacy of therapy by mismatching the killer cell immunoglobulin-like receptors present on NK cells with the human leukocyte antigen ligands of tumor cells, promoting graft-versus-tumor effects [114]. When activated, CAR-NK cells also secrete a more favorable cytokine profile compared to CAR-T cells, which reduces the risk of patients developing CRS [113]. Research has already begun to assess the efficacy of CAR-NK cells in GBM therapy.

### 4.1. EGFRvIII 

Murakami et al. created CAR-NK cells targeting EGFRvIII present on the surface of GBMs. The CAR construct consisted of an scFv responsible for the recognition of EGFRvIII, the CD3ζ signaling domain and the CD28 and 4-1BB co-stimulatory domains. The KHYG-1 cell line to which the CAR construct was transduced was used as the source of NK cells. The anti-EGFRvIII CAR-NK cells created in this way were assessed for anticancer activity in vitro. For this purpose, CAR-NK cells were co-cultured with GBM cell lines—U87MG and U87MG constitutively expressing EGFRvIII (U87-EGFRvIII). It was estimated that anti-EGFRvIII CAR-NK cells induced apoptosis at a level of 78.2 ± 1.6% in the case of U87-EGFRvIII at an effector to target ratio of 5:1. In comparison, under the same conditions, control NK cells induced apoptosis at a level of 29.7 ± 1.3% [115].

Ma et al. examined the efficacy of the synergistic therapy using anti-EGFR CAR-NK cells and an oncolytic virus expressing Il-15/Il-15Rα (OV-IL15C). The CAR-NK cells used were capable of recognizing both EGFR and EGFRvIII. In in vitro studies, the antitumor effectiveness of anti-EGFR CAR-NK cells against EGFR^+^ and EGFRvIII^+^ cell lines was assessed. Anti-EGFR CAR-NK cells effectively killed target cells while secreting increased amounts of IFN-γ and TNF-α. In an in vivo study in an orthotopic GBM model, mice treated with anti-EGFR CAR-NK cells achieved increased survival compared to the control group. The authors then assessed the efficacy of the synergistic therapy of anti-EGFR CAR-NK cells with OV-IL15C compared to anti-EGFR CAR-NK cells with oncovirus controls and to anti-EGFR CAR-NK cells or OV-IL15C alone. The mice treated with both the synergistic therapy and each of the aforementioned drugs alone showed reduced tumor growth rate and prolonged survival. However, the longest survival was observed in the mice treated simultaneously with anti-EGFR CAR-NK cells and OV-IL15C, which indicates that these drugs have a synergistic effect. Additionally, the authors demonstrated in an immunocompetent mouse model that the presence of OV-IL15C prolongs the survival of CAR-NK cells and also reduces their cell exhaustion [116].

### 4.2. HER2

In one study, the authors used the NK-92 cell line and transduced these cells with a HER2-specific CAR also containing CD28 and CD3ζ domains (NK-92/5.28.z). The cells constructed in this way effectively killed HER2^+^ GBM cells, while their activity against HER2^-^ cells was minimal and comparable to the original NK cells. Cell activity was then assessed in vivo, first on orthotopic GBM xenografts in NSG mice and then on immunocompetent mice models. In the first case, LN-319 cells were injected into mice’s brains, followed 7 days later by the intratumoral NK-92/5.28.z infusion. This resulted in the inhibition of tumor growth as well as prolonged survival time of the treated mice (median survival 200.5 days) compared to the mice treated with non-specific NK-92 cells (median survival 74.5 days). In an immunocompetent mouse model of GBM, out of eight mice treated with NK-92/5.28.z, five of them rejected tumor cells. On Day 126 of the experiment, these mice were re-implanted with GBM cells in the contralateral brain hemisphere, but they were not treated. Despite this, the mice rejected the tumor cells again and survived until the end of the experiment (Day 194). These findings suggest that NK-92/5.28.z therapy results in long-term immunological memory [117].

Strecker et al. also created HER2-specific NK-92/5.28.z (anti-HER2.CAR/NK-92) and confirmed the antitumor activity in vitro on several cell lines. The authors observed in vitro that after the administration of anti-HER2.CAR/NK-92 cells, surrounding cells representing TME cells responded with the increased expression of pro-inflammatory cytokines such as IL-6 and IL-8. Additionally, researchers found the upregulation of the PD-L1 on GBM cells and other cells of the TME in the presence of anti-HER2.CAR/NK-92 cells. Therefore, it was decided to investigate the efficacy of the therapy using anti-HER2.CAR/NK-92 cells and anti-PD-1 immunoadhesin via a targeted adeno-associated viral vector. In in vivo models, this combination resulted in complete tumor resolution in some of the treated mice and significantly extended survival time [118].

### 4.3. Others

Zuo et al. conducted a preclinical study assessing the efficacy of second-generation CAR-NK cells, created from the NK-92 cell line, directed against the GD2 antigen (GD2-CAR-NK-92) in the treatment of DIPG occurring in children. In vivo studies in orthotopic DIPG xenograft mouse models with high GD2 expression (TT150630 DIPG mice) and low GD2 expression (TT190326 DIPG mice) showed that GD2-CAR-NK-92 cells used in TT150630 DIPG mice exhibited a strong anti-tumor response and prolonged the survival of the animals. A total of 50% of mice treated with GD2-CAR-NK-92 cells survived more than 40 days, while all mice in the control group died after 20 days. When GD2-CAR-NK-92 cells were used in the treatment of TT190326 DIPG mice, such a significant difference was not achieved—all mice from the control group died after approximately 90 days, while all mice treated with GD2-CAR-NK-92 cells died after approximately 100 days [119]. It should be noted, however, that the therapy with CAR-NK cells generated from the NK-92 line may be limited by the need to irradiate them before their administration to the patient, which may result in a reduction and shortening of their in vivo viability [119].

In the immunosuppressive TME present in GBM, there is a local high concentration of transforming growth factor beta (TGF-β) that limits the effector functions of CAR-NK cells. Chaudhry et al. showed that the treatment with CAR-NK cells directed against the B7-H3 antigen (B7-H3 CAR-NK) with the immunosuppressive TGF-β resulted in a reduction in tumor cell lysis by approximately 28% (percentage of lysis without the presence of TGF-β: 89.73 ± 2.44% vs. lysis in the presence of TGF-β: 61.75 ± 3.42%; at E:T = 20:1). Additionally, it has been confirmed that TGF-β reduces the expression of cluster of differentiation 16 (CD16) and NKG2D activating receptors on CAR-NK cells, thereby reducing their ability to be activated in the absence of the CAR target antigen [114]. Therefore, it seems important to introduce a modification to CAR-NK cells that would make them resistant to TGF-β-induced immunosuppressive signaling. Burga et al. presented three different ways in which the impact of negative TGF-β signaling on the effector functions of CAR-NK cells can be reduced. The first is the dominant negative TGF-β receptor (DNR), which abolishes TGF-β signaling by blocking the phosphorylation of mothers against decapentaplegic homolog 2 and 3, necessary for the initiation of the transcription of immunosuppressive genes. The remaining two constructs contained a truncated TGFβ receptor II domain fused to a DNAX-activation protein 12 activating motif (NKA construct) or to a synthetic Notch-like receptor fused to p65 (NKCT construct), respectively. The NKA and NKCT constructs, like DNR, bind TGF-β present in the TME, but unlike DNR, which only inhibits the transmission of an immunosuppressive signal, NKA and NKCT additionally change the immunosuppressive signal into a pro-inflammatory signal by activating the transcription of genes associated with the cytotoxic phenotype of CAR-NK cells [120]. Chaudhry et al. therefore decided to evaluate the effector functions of B7-H3 CAR-NK cells, which additionally co-express DNR. It was shown that such a modification limited the TGF-β-dependent reduction in the expression of CD16 and NKG2D receptors, and also allowed for the preservation of the effector functions of CAR-NK cells in the presence of TGF-β [114].

## 5. Limitations of CAR-Expressing Cells

One of the main limitations affecting the efficacy of immunotherapy is the TME. There are some TME-related challenges that require further research [121]. The heterogeneity of tumor cells contributes to the diversity of the expression of their surface antigens. The selection of a single CAR target antigen may, therefore, be associated with therapy resistance related to the presence of tumor cells that do not express the CAR target antigen and hence do not trigger a CAR-mediated antitumor cell response [122]. Verhaak et al., thanks to the use of genomic profiling techniques, identified four molecular subtypes of GBM, namely, proneural, neural, classical and mesenchymal, differing in gene expression and, therefore, in the expression of antigens occurring on the surface of cancer cells. For example, the classical subtype was characterized by a high level of EGFR amplification compared to other subtypes, and the EGFRvIII mutation was most often found in the classical subtype [123]. The key to effective tumor cell elimination via CAR-expressing cells is the identification of the correct target antigen. Because of the differences among molecular subtypes of GBM, there is a need to match the target antigen for the CAR to the surface antigens that occur on cancer cells in a given subtype. The construction of Dual-CAR cells (cells that express two different types of CARs targeting two different antigens) or a bispecific tandem CAR (one type of CAR with two binding domains targeting two different antigens) or even trivalent CAR-T cells (which co-express three independent types of CARs) with target antigens selected to overlap as much as possible with the expression of antigens on cancer cells of particular molecular subtypes could increase the effectiveness of the therapy, because this type of construction would limit the antigen escape mechanism during which cancer cells may reduce the expression of the CAR target antigen on their surface [17,66,122].

Determining the molecular subtype of a given patient’s cancer could not only allow for better personalization of therapy for a given patient, but also serve as a predictor of the effectiveness of this form of immunotherapy. Verhaak et al. showed that aggressive treatment (concomitant chemotherapy and radiotherapy or more than three consecutive cycles of chemotherapy) reduced mortality in the classic and mesenchymal subtypes of GBM, but had no effect on survival in the proneural subtype [123]. Jakovlevs et al. in their work indicated the possibility of identifying individual molecular subtypes of GBM through immunohistochemical testing, which is cheaper and more easily available compared to the method based on gene expression testing. In the case of selecting personalized therapy, it is necessary to perform a number of diagnostic tests; therefore, reducing the costs of the procedure for identifying patients eligible for therapy is extremely important [124].

It is worth noting that cancer cells are one of the components that create the TME, and the differences between individual molecular subtypes of GBM cells may result in the creation of a TME with a different molecular profile that will model the effectiveness of CAR-expressing cells in a different way. For example, abnormal angiogenesis caused by the uncontrolled secretion of factors promoting vascular development, among others, by tumor cells causes circulation disorders within the tumor, on the one hand, thus reducing the infiltration of the TME by CAR-expressing cells, and on the other hand, contributing to the creation of the immunosuppressive TME by hypoxia of the areas affected by the tumor process [6,122,125]. The reduced local oxygen availability favors the transition of cells present in the TME to anaerobic metabolism, thus leading to the acidification of the local environment [126]. It should be noted, however, that the unfavorable immunosuppressive TME is mainly responsible for the tumor-infiltrating immunosuppressive cells of the immune system (Treg lymphocytes, myeloid-derived suppressor cells, macrophages and neutrophils with the M2 phenotype), attracted to the tumor site by chemoattractants secreted by tumor cells. The immunosuppressive infiltrate, in turn, contributes to the abnormal functioning of CAR-expressing cells by secreting immunosuppressive cytokines or expressing checkpoint inhibitors, which ultimately leads to CAR-expressing cells entering a state of exhaustion or anergy [6,121,126,127]—see Figure 3.

The results of preclinical and clinical studies indicate a certain potential of cells expressing CAR in extending the survival time of patients with HGG. However, in order to strengthen the effector mechanisms of cells expressing CAR, it is necessary to conduct further research on the optimization of the CAR construct as well as the cells themselves by introducing transgenes into them in order to improve their survival in the immunosuppressive TME. Efforts should also be made to reprogram the immunosuppressive TME into a pro-inflammatory TME, which will contribute to improving the functioning of CAR-expressing cells in the TME [128,129]. It is also worth directing further work to better characterize the interactions between individual molecular subtypes of HGG and the TME. Understanding the molecular pathways contributing to the development of resistance to therapy with CAR-expressing cells will enable more effective counteraction [123,124].

## 6. Conclusions

Despite the promising results of preclinical studies, most of the patients described in the analyzed clinical trials who received the CAR-expressing cell preparation died of cancer. However, transient effectiveness of CAR-expressing cells was observed, resulting in a reduction in tumor mass and prolonging of patient survival compared to the median survival time of patients with HGG. It is therefore necessary to conduct further work on the CAR construct in order to increase the effectiveness of CAR cells, as well as to consider synergistic therapies combining the administration of CAR cells with agents that improve their effectiveness. A clear determination of whether CAR cells will become one of the therapeutic procedures used in HGG will only be possible after conducting more clinical trials on larger groups of patients. It should also be noted that one of the reasons limiting the effectiveness of CAR therapy is the immunosuppressive TME. Therefore, in order to increase the effectiveness of HGG therapy, it is necessary to conduct further research on the TME. Only a complete understanding of the interactions between cancer cells and their environment, taking into account the molecular and genetic mechanisms that shape the individual image of the TME in each patient, can enable the development of more effective, personalized therapies.

## Figures and Tables

**Figure 1 cancers-16-00623-f001:**
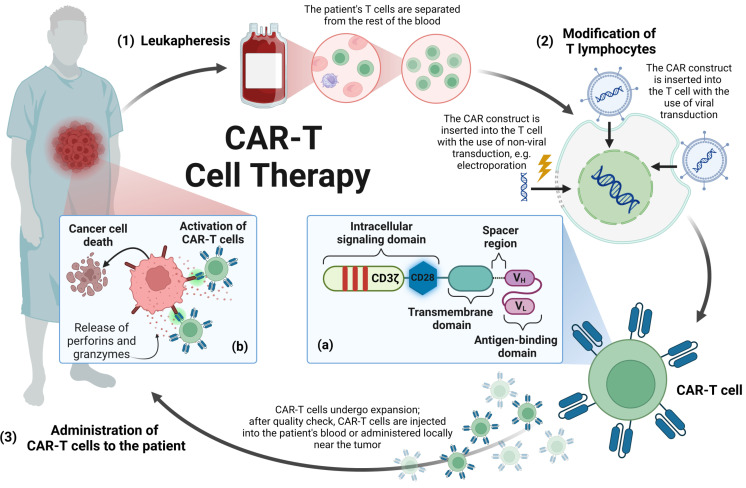
Manufacturing process of CAR-T cell therapy: the manufacturing process of CAR-T cell therapy begins with (1) leukapheresis during which T cells are isolated from the patient’s blood. (2) Then, T cells are transduced with the CAR-encoding viral vector or by non-viral methods, e.g., electroporation. (3) After expansion, CAR-T cells pass quality check. They are cryopreserved and, in this form, CAR-T cells are transported to a medical facility where they will be administered to the patient. (**a**) The CAR construct is composed of the extracellular antigen-binding domain, the spacer region, the transmembrane domain and the intracellular signaling domain. The intracellular signaling domain consists of a CD3ζ cytoplasmic domain with three immunoreceptor tyrosine-based activation motifs and may contain a co-stimulatory domain, e.g., CD28 and a transcription factor enabling the production of pro-inflammatory cytokines. (**b**) Effector mechanism of CAR-T cells: Binding the CAR to its target antigen present on the cancer cell triggers a signaling cascade ultimately leading to the activation of the CAR-T cell. Activated CAR-T cells lead to the death of cancer cells by secreting cytokines or granzymes, or interacting via the Fas and FasL pathway. CAR-T—T lymphocytes expressing chimeric antigen receptor, CD3ζ—T-cell receptor T3 zeta chain, CD28—cluster of differentiation 28, V_H_—heavy chain variable segment, V_L_—light chain variable segment. Image created with biorender.com (accessed on 8 November 2023).

**Figure 2 cancers-16-00623-f002:**
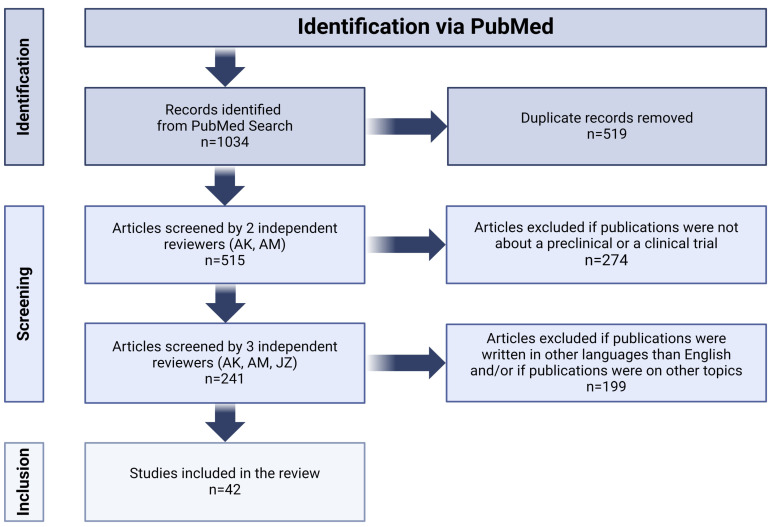
Flow diagram of systematic review according to PRISMA guidelines. Image created with biorender.com (accessed on 26 January 2024).

**Figure 3 cancers-16-00623-f003:**
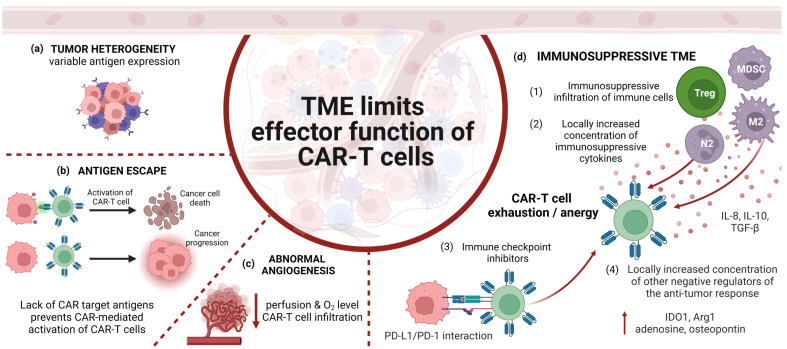
The tumor microenvironment limits the functioning of CAR-T cells by (**a**) heterogeneity of cancer cells that make up the tumor, which is associated with different expression of antigens on their surface; therefore, CAR-T cells can eliminate only part of the population of cancer cells expressing the CAR target antigen.; (**b**) antigen escape—cancer cells during therapy may reduce the expression of the CAR target antigen, and the lack of CAR target antigens prevents CAR-mediated activation of CAR-T cells; (**c**) abnormal angiogenesis, which reduces the efficiency of the infiltration of CAR-T cells to the tumor, their effector functions being additionally inhibited by hypoxic conditions within the TME; (**d**) immunosuppressive conditions in the TME caused by the immunosuppressive infiltration of immune cells (e.g., Tregs, MDSCs, macrophages with the M2 phenotype, neutrophils with the N2 phenotype); the aforementioned cells contribute to the locally increased concentration of immunosuppressive cytokines (e.g., IL-8, IL -10, TGF-β) and increase the pool of cells with the high expression of checkpoint inhibitors (e.g., PD-1L). Additionally, in the TME there is a locally increased concentration of other negative regulators of the anti-tumor response (e.g., IDO1, Arg1, adenosine, osteopontin). This leads to the disturbed metabolism of CAR-T cells, weakening their effector functions and promoting the entry of CAR-T cells into a state of anergy or exhaustion. CAR-T—T lymphocytes expressing chimeric antigen receptor, TME—tumor microenvironment, PD-1—programmed death receptor 1, PD-L1—programmed death-ligand 1, Treg—regulatory T cell, myeloid-derived suppressor cells, M2—M2 phenotype macrophages, N2—N2 phenotype neutrophiles, IL-8—interleukin 8, IL-10—interleukin 10, TGF-β—transforming growth factor beta, IDO1—indoleamine 2,3-dioxygenase-1, Arg1—arginase 1. Image created with biorender.com (accessed on 8 November 2023).

**Table 1 cancers-16-00623-t001:** Characteristics of HGGs.

Type of Cancer		Molecular Markers of Cancer	Morphological Characteristics of the Tumor	Survival Rate	References
pHGG	Diffuse midline glioma, H3 K27-altered	H3-3A, HIST1H3B, HIST1H3BC TP53, ACVR1, PDGFRA, EGFR, EZHIP	Diffuse infiltrative growth, affecting midline structures. Located in the brain stem, thalamus and spinal cord.	10.5–19.6 months	[7,8,9,10,11]
	Diffuse hemispheric glioma, H3 G34-mutant	H3-3A, TP53, ATRX, MGMT	Glioblastoma-like or PNET-like histomorphology. Mostly located in cerebral hemispheres.	23.5 months	[7,8,10,12]
	Diffuse pediatric-type HGG, H3-wildtype and IDH-wildtype	H3-wildtype (H3-3A, HIST1H3B, HIST1H3BC), IDH-wildtype (IDH1, IDH2) PDGFRA, MYCN, EGFR	Located supratentorial, in the brain stem or cerebellum.	17 months	[7,8,10,13]
	Infant-type hemispheric glioma	NTRK 1/2/3, ALK, ROS, MET	Located in cerebral hemispheres.	-	[7,8,10,14]
aHGG	Astrocytoma, IDH-mutant, CNS WHO grade 4	IDH1, IDH2 ATRX, TP53	Necrosis and/or microvascular proliferation, oligodendroglioma-like components.	14–18 months	[9,10,15]
	Glioblastoma, IDH-wildtype, CNS WHO grade 4	IDH-wildtype (IDH1/2) TERT, EGFR amplification, +7/−10 chromosome copy-number changes	Necrosis and/or microvascular proliferation.	14–18 months	[9,10,15]

HGG—high-grade glioma, pHGG—pediatric high-grade glioma, aHGG—adult high-grade glioma, CNS—Central Nervous System, WHO—World Health Organization, H3 K27—the 27th amino acid in Histone H3, H3-3A—H3.3 Histone A, HIST1H3B—histone cluster 1 H3b, HIST1H3BC—histone cluster 1 H3bc, TP53—cellular tumor antigen p53, ACVR1—activin A receptor type I, PDGFRA—platelet-derived growth factor receptor A, EGFR—epidermal growth factor receptor, EZHIP—uncharacterized protein CXorf67, H3-G34—glycine to arginine or valine substitutions at codon 35, ARTX—X-linked helicase II, MGMT—methylated-DNA–protein-cysteine methyltransferase, IDH1—isocitrate dehydrogenase 1, IDH2—isocitrate dehydrogenase 2, MYCN—basic helix–loop–helix transcription factor, NTRK 1—neurotrophic receptor tyrosine kinase 1, NTRK 2—neurotrophic receptor tyrosine kinase 2, NTRK 3—neurotrophic receptor tyrosine kinase 3, ALK—ALK receptor tyrosine kinase, ROS—proto-oncogene, receptor tyrosine kinase L homeolog, MET—proto-oncogene, receptor tyrosine kinase, TERT—telomerase reverse transcriptase.

**Table 2 cancers-16-00623-t002:** Currently recruiting clinical trials testing the use of CAR-T cell therapy in pHGG and aHGG.

Drug	ClinicalTrials.gov Identifier	Phase of Clinical Study	Estimated Number of Patients	Studied Patient Population	Method of Administration	Dosage	Reference
Anti-GD2 CAR-T cells	NCT05544526	Phase 1	12	Patients with DMG, up to 16 y.o.	-	-	[92]
Anti-GD2 CAR-T cells	NCT04196413	Phase 1	54	Patients with H3K27M-mutated DIPG, 2–50 y.o.	Intravenously; Intracerebroventricularly	3 × 10^5^–3 × 10^6^ cells/kg; 10 × 10^6^–100 × 10^6^ cells	[93]
GD2.C7R-T cells	NCT04099797	Phase 1	34	Patients with GD2-expressing newly diagnosed DMG, 1–21 y.o.	Intravenously; Intracerebroventricularly	3 × 10^6^ cells/m^2^; 5 × 10^6^–5 × 10^7^ cells	[94]
B7-H3-CAR-T cells	NCT05835687	Phase 1	36	Patients with primary CNS tumor, up to 21 y.o.	Locoregionally	-	[95]
B7-H3-specific CAR-T cells	NCT04185038	Phase 1	90	Patients with refractory or recurrent CNS tumor or with DIPG or DMG, 1–26 y.o.	Locoregionally	-	[35]
CAR.B7-H3T cells	NCT05366179	Phase 1	36	Patients with recurrent supratentorial or infratentorial GBM, above 18 y.o.	Intraventricularly	2–5 × 10^6^ cells/infusion	[96]
B7-H3 CAR-T cells	NCT04077866	Phase 1, Phase 2	40	Patients with relapsed refractory B7-H3^+^ GBM, 18–75 y.o.	Intratumorally or intracerebroventricularly	-	[97]
B7-H3 CAR-T	NCT05474378	Phase 1	39	Patients with recurrent/progressive HGG, above 18 y.o.	Locoregionally	5–100 × 10^6^ cells/dose	[98]
B7-H3 CAR-T	NCT04385173	Phase 1	12	Patients with relapsed/refractory B7-H3^+^ grade IV GBM, 18–75 y.o.	Intratumorally or intracerebroventricularly	-	[99]
Anti-IL-13Rα2 UCAR-T cells; Anti-B7-H3 UCAR-T cells	NCT05752877	Not applicable	12	Patients with advanced, locally advanced or recurrent glioma, 18–70 y.o.	Locoregionally	1–5 × 10^7^ cells	[100]
Anti-IL-13Rα2 CAR-T cells	NCT05540873	Phase 1	18	Patients with HGG, 19–74 y.o.	Intravenously	-	[101]
Anti-IL-13Rα2 CAR-T cells	NCT04661384	Phase 1	30	Patients with leptomeningeal GBM, ependymoma or medulloblastoma, above 18 y.o.	Intracerebroventricularly	-	[102]
Anti-IL-13Rα2 CAR-T cells with or without nivolumab and ipilimumab	NCT04003649	Phase 1	60	Patients with grade IV GBM, above 18 y.o.	Locoregionally	-	[103]
Chlorotoxin (EQ)-CD28-CD3ζ-CD19t-expressing CAR-T lymphocytes	NCT04214392	Phase 1	36	Patients with MMP-2^+^ recurrent or progressive GBM, above 18 y.o.	Intracavitary/intratumorally and intraventricularly	-	[104]
CHM-1101 CAR-T cells	NCT05627323	Phase 1	42	Patients with MMP-2^+^ recurrent or progressive GBM, above 18 y.o.	Intracavitary/intratumorally and intraventricularly	-	[105]
SNC-109 CAR-T cells	NCT05868083	Phase 1	16	Patients with recurrent GBM, 18–70 y.o.	-	2 × 10^4^ cells	[106]
Anti-HER2 CAR-T cells	NCT03500991	Phase 1	48	Patients with HER2^+^ CNS tumor, 1–26 y.o.	Locoregionally	-	[107]
NKG2D-based CAR-T cells	NCT05131763	Phase 1	3	Patients with NKG2DL^+^ tumors including CNS tumors, 18–75 y.o.	Intravenously	1–10 × 10^6^ cells/kg	[108]
IL-8R modified CD70 CAR-T cells	NCT05353530	Phase 1	18	Patients with CD70^+^ MGMT-unmethylated GBM, 18–80 y.o.	Intravenously	1 × 10^6^–1 × 10^8^ cells/kg	[109]
SC-CAR4BRAIN (T cells expressing B7-H3, EGFR806, HER2 and IL13-zetakine CARs	NCT05768880	Phase 1	72	Patients with DIPG, DMG or refractory/recurrent CNS tumor, 1–26 y.o.	Intravenously	-	[110]

CAR-T—T lymphocytes expressing chimeric antigen receptor, pHGG—pediatric high-grade glioma, aHGG—adult high-grade glioma, GD2—disialoganglioside, DMG—diffuse midline glioma, y.o.—years old, H3K27M—substitution of lysine 27 to methionine in histone H3, DIPG—diffuse intrinsic pontine glioma, B7-H3—B7 homologue 3 protein, CNS—central nervous system, GBM—glioblastoma, IL-13Rα2—interleukin-13 receptor subunit α-2, MMP-2—matrix metalloproteinase-2, HER2—human epidermal growth factor receptor 2, NKG2D—natural killer group 2 member D protein, NKG2DL—natural killer group 2 member D protein ligand, CD70—cluster of differentiation 70, MGMT—O^6^-methylguanine-DNA methyltransferase, EGFR—epidermal growth factor receptor.

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
