# Peer review of "Chimeric Antigen Receptor T Cell and Chimeric Antigen Receptor NK Cell Therapy in Pediatric and Adult High-Grade Glioma—Recent Advances"

_cancers, 2024, doi:10.3390/cancers16030623_

Round 1

Reviewer 1 Report

Comments and Suggestions for Authors

There are several reviews written in recent years that deal with the topic of CAR-T therapy in glioblastoma. However, in this review, the role of certain target antigens such as GD2, B7-H3, EphA2, IL-13Rα2, or EGFRvIII is analyzed clearly, highlighting pre-clinical and clinical studies, as well as limitations of CAR-expressing cells. This makes the review interesting and fairly original.

Author Response

Dear Sir or Madam, thank you very much for the review of our manuscript entitled: “CAR-T and CAR-NK Cell Therapy in Pediatric and Adult High-Grade Glioma—Recent Advances”

Comment

There are several reviews written in recent years that deal with the topic of CAR-T therapy in glioblastoma. However, in this review, the role of certain target antigens such as GD2, B7-H3, EphA2, IL-13Rα2, or EGFRvIII is analyzed clearly, highlighting pre-clinical and clinical studies, as well as limitations of CAR-expressing cells. This makes the review interesting and fairly original.

Our comment

In response to your comment, we would like to thank you for appreciating our manuscript.

Reviewer 2 Report

Comments and Suggestions for Authors

The manuscript "CAR-T and CAR-NK cell therapy in pediatric and adult high-grade glioma – recent advances" represents a comprehensive, non-systematic review on the use of immune cells, expressing a chimeric antigen receptor, in the treatment of high-grade glial tumours. Authors have structured the information by the type of immune cells (CAR-T vs CAR-NK) and age of patients (paediatric vs adult gliomas), further stratifying the data by target antigens and preclinical vs clinical studies. Thus, the structure of the article is logical and clear. The description is extensive and remarkably detailed. Thus, the article can serve as a reference for planning the further studies. The manuscript is appropriately illustrated.

Considering the dismal prognosis of adult glioblastoma and paediatric diffuse midline glioma, the topic undoubtedly is timely and important. The contents of the manuscript correspond to the scope of the journal “Cancers”, the section “Tumor microenvironment” and special issue “Targeting the Tumor Microenvironment Volume II”. The authors refer to studies, performed with up-to-dated technologies. The level of English language is generally good, although occasional misprints should be corrected, as further indicated.

Few corrections would be recommended, please:

1.      High-grades gliomas in children and in adults represent very different entities. I would highly advise to provide short (!) characteristics of the corresponding tumours (e.g., glioblastoma vs. IDH-mutant astrocytoma, grade 4 vs diffuse midline glioma) in accordance with the current WHO classification: e.g., morphological and molecular diagnostic criteria and survival (this parameter is particularly important to assess the results of experimental treatment).

2.      Please, note, that complete tumour resection is hardly possible in high-grade glioma.

3.      Indicate, please, the author or source of the illustrations (i.e., if the figures are original, or reprinted with permission, or used in accordance with Creative Commons licence, etc.).

4.      Please, always indicate the pharmacological group of the described medications, e.g. tocilizumab (line 155), anakinra (line 158), situximab (line 608), etc., throughout the article.

5.      In glioblastoma, several molecular subtypes have been identified both by gene expression and immunohistochemistry. These subtypes were shown to predict the efficacy of the conventional treatment. See, please, Verhaak et al., 2010 (PMID: 20129251) and Jakovlevs et al., 2019 (PMID: 32146793) for gene expression-based and immunohistochemical subtyping, respectively. Molecular subtyping, esp. by immunohistochemistry, is easily available in diagnostic surgical pathology labs. Please, discuss if these molecular subtypes could serve as predictors for the efficiency of CAR cell-mediated immunotherapy?

6.      Although the level of English is reasonably high, few minor misprints should be corrected, e.g., “squamos” (line 201); “fidderentation” (line 712), et al.

7.      Please, format the references according to the Instructions for Authors, as indicated by “Cancers”.

Finally, I would like to thank the authors for their contribution. It was a pleasure and a true honour to review this manuscript.

Comments on the Quality of English Language

Although the level of English is reasonably high, few minor misprints should be corrected, e.g., “squamos” (line 201); “fidderentation” (line 712), et al.

Author Response

Dear Sir or Madam, thank you very much for the review of our manuscript entitled: “CAR-T and CAR-NK Cell Therapy in Pediatric and Adult High-Grade Glioma—Recent Advances”

In response to your comments, we would like to thank you so much for appreciating our manuscript. We found your comments extremely valuable and thanks to them we were able to improve our manuscript.

The manuscript has been marked up using the “Track Changes”.

Comment 1

High-grades gliomas in children and in adults represent very different entities. I would highly advise to provide short (!) characteristics of the corresponding tumours (e.g., glioblastoma vs. IDH-mutant astrocytoma, grade 4 vs diffuse midline glioma) in accordance with the current WHO classification: e.g., morphological and molecular diagnostic criteria and survival (this parameter is particularly important to assess the results of experimental treatment).

Revision and our comment

Characteristics of high-grade gliomas have been added in the form of a table to the manuscript (page 2).

Comment 2

Please, note, that complete tumour resection is hardly possible in high-grade glioma (Tabele 1 on page …)

Revision and our comment

We have updated the manuscript with this information (lines 70-72).

Comment 3

Indicate, please, the author or source of the illustrations (i.e., if the figures are original, or reprinted with permission, or used in accordance with Creative Commons licence, etc.).

Revision and our comment

The illustrations are original and were made in the BioRender program by Julia Zarychta (one of the authors of the work).

Comment 4

Please, always indicate the pharmacological group of the described medications, e.g. tocilizumab (line 155), anakinra (line 158), situximab (line 608), etc., throughout the article.

Revision and our comment

The pharmacological group of the described medications has been added:

  • tocilizumab (line 192-193) – humanized monoclonal antibody against the interleukin-6 receptor
  • anakinra (line 196-197) – human interleukin 1 receptor antagonist protein
  • temozolimide (line 649) – alkylating agent
  • siltuximab (line 653-654) – chimeric monoclonal antibody against the interleukin-6
  • cyclophosphamide (line 679-680) – antineoplastic agent metabolized to active alkylating metabolites
  • fludarabine (line 680-681) – purine analogue
  • bevacizumab (line 694) – humanized monoclonal antibody against VEGF-A

Comment 5

In glioblastoma, several molecular subtypes have been identified both by gene expression and immunohistochemistry. These subtypes were shown to predict the efficacy of the conventional treatment. See, please, Verhaak et al., 2010 (PMID: 20129251) and Jakovlevs et al., 2019 (PMID: 32146793) for gene expression-based and immunohistochemical subtyping, respectively. Molecular subtyping, esp. by immunohistochemistry, is easily available in diagnostic surgical pathology labs. Please, discuss if these molecular subtypes could serve as predictors for the efficiency of CAR cell-mediated immunotherapy?

Revision and our comment

The aforementioned works were added to the manuscript, and the emphasis was put on the possibility of using GBM molecular subtypes as predictors of the response to CAR-T/CAR-NK therapy (lines 901 - 938, lines 985 - 989).

Comment 6

Although the level of English is reasonably high, few minor misprints should be corrected, e.g., “squamos” (line 201); “fidderentation” (line 712), et al.

Revision and our comment

The work has been re-checked for correct English and spelling. Misprints were corrected:

  • “squamos” into “squamous” – line 239,
  • “fidderentation” into “differentiation” – line 761,
  • “immunocompletent” into “immunocompetent” – line 824, line 831.

Comment 7

Please, format the references according to the Instructions for Authors, as indicated by “Cancers”.

Revision and our comment

The references have been corrected according to the instructions of "Cancers".

Reviewer 3 Report

Comments and Suggestions for Authors

Dear Authors, 

This manuscript titled CAR-T and CAR-NK Cell Therapy in Pediatric and Adult 2 High-Grade Glioma—Recent Advances is well organized and presented.

The Authors discussed promising results of preclinical studies and patients described in 948 the analyzed clinical trials, who received the CAR-expressing cell preparation. This paper is very informative and laborious. All references are appropriate and adequate to the subject.  

I suggest to add Prisma graph and more methodology, maybe also statistic about increased number of papers  during the last 10 years .   

Thank you

Author Response

Dear Sir or Madam, thank you very much for the review of our manuscript entitled: “CAR-T and CAR-NK Cell Therapy in Pediatric and Adult High-Grade Glioma—Recent Advances”.

We would like to thank you for appreciating our manuscript and providing us with your comments, which we found extremely valuable and helpful in improving our manuscript.

The manuscript has been marked up using the “Track Changes”.

Comment 1

This manuscript titled CAR-T and CAR-NK Cell Therapy in Pediatric and Adult 2 High-Grade Glioma—Recent Advances is well organized and presented.The Authors discussed promising results of preclinical studies and patients described in 948 the analyzed clinical trials, who received the CAR-expressing cell preparation. This paper is very informative and laborious. All references are appropriate and adequate to the subject. 

I suggest to add Prisma graph and more methodology, maybe also statistic about increased number of papers  during the last 10 years.  

Revision and our comment

Methodology description and the graph have been added to the manuscript (lines 107 - 122).